# Investigation of Extracted Plasma Cell-Free DNA as a Biomarker in Foals with Sepsis

**DOI:** 10.3390/vetsci11080346

**Published:** 2024-08-01

**Authors:** Kallie J. Hobbs, Bethanie L. Cooper, Katarzyna Dembek, M. Katie Sheats

**Affiliations:** Department of Clinical Sciences, College of Veterinary Medicine, North Carolina State University, Raleigh, NC 27695, USA; kjhobbs@ncsu.edu (K.J.H.); bplewis2@ncsu.edu (B.L.C.); kdembek@ncsu.edu (K.D.)

**Keywords:** foals, neutrophils, horses, cell-free DNA

## Abstract

**Simple Summary:**

Cell-free DNA (cfDNA) are pieces of DNA released from cells into body fluids. Previous studies in adult horses found that plasma cfDNA concentrations differed significantly between healthy horses and those with emergencies like colic. These studies also showed that accurate measurement of plasma cfDNA in adult horses required extraction, due to matrix effects of equine plasma. It’s unclear if similar issues exist in foal plasma. In our study, we aimed to determine if foal plasma has similar interference, and if there are differences in cfDNA levels between healthy, sick non-septic (SNS), and septic foals. Cell-free DNA was measured directly in plasma and after extracting it using a kit in 60 foals. Direct measurement of cfDNA in foal plasma was found to be inaccurate due to matrix effect. However, even after cfDNA extraction, there were no significant differences in plasma cfDNA levels or the ratio of cfDNA to neutrophils between healthy foals, SNS foals, and septic foals. Future research should focus on understanding how neutrophils function during foal sepsis.

**Abstract:**

Cell-free DNA (cfDNA) is fragmented extracellular DNA that is released from cells into various body fluids. Previously published data from adult horses supports cfDNA as a potential disease biomarker, but also shows that direct measurement in plasma is inaccurate due to matrix effect. It is currently unknown whether a similar matrix effect exists in foal plasma. Given this, the objectives of the current study were to investigate foal plasma for potential matrix effect during fluorescence measurement of cfDNA using a Qubit fluorometer, and to determine whether neat and/or extracted plasma cfDNA concentrations are significantly different in healthy, sick non-septic (SNS) or septic foals. We hypothesized that matrix effect would interfere with accurate fluorescent measurement of cfDNA in foal plasma. Further, we hypothesized that mean extracted cfDNA concentrations, and/or extracted cfDNA:neutrophil ratio, would be elevated in plasma of septic foals compared to healthy or SNS foals. Cell-free DNA was measured in neat plasma, and following DNA extraction with a commercial kit, from 60 foals. Foal plasma exhibited both autofluorescence and non-specific dye binding, confirming matrix effect. However, even with extraction, no significant difference was found in cfDNA concentrations, or cfDNA:neutrophil ratios, between healthy (sepsis score ≤ 5), SNS (sepsis score 6–11 and negative blood culture), or septic (sepsis score ≥ 12 ± positive blood culture) foals. Our data show that matrix effect interferes with accurate Qubit measurement of cfDNA in foal plasma and supports previous findings that plasma cfDNA concentrations are not associated with sepsis diagnosis in foals. Further research is needed to better understand neutrophil function and dysfunction in foal sepsis.

## 1. Introduction

In foals, sepsis is a serious health concern that can have mortality rates as high as 40–60% [1]. There has been progress in recent years regarding foal sepsis diagnosis and prognosis; however, additional work is needed to better understand the immunopathology of sepsis in foals, and to identify additional biomarkers and novel therapeutic targets. In humans with sepsis, including infants [2], cell-free DNA (cfDNA) is a biomarker that has exhibited prognostic value, with higher levels of cfDNA being correlated with increased mortality [3]. Cell-free DNA is fragmented extracellular DNA that is released into body fluids such as plasma, as a result of necrosis, apoptosis, and/or extrusion of extracellular traps from cells like neutrophils (i.e., NETs) [4,5]. NETosis is one method neutrophils use to kill bacteria, trapping invading organisms in extruded strands of DNA intercalated with destructive enzymes such as neutrophil elastase and myeloperoxidase. While this function is designed for host defense, dysregulated NETosis can lead to thrombosis, inflammation, and multiple organ failure in patients with sepsis and has been correlated with disease severity [6]. Increased cfDNA is also associated with systemic inflammation and disease severity in humans with different types of liver injury [7], in part through the deleterious effects of exaggerated neutrophil recruitment and activation. Interestingly, in a mouse model of liver injury, intravenous administration of DNase I decreased cfDNA and significantly decreased liver injury [7]. Similarly, administration of DNase or a NETosis inhibitor (PAD4 inhibitor) in an infant mouse model of sepsis decreased sepsis severity [2]. These data show cfDNA is not only a potential biomarker, but also a potential therapeutic target.

In equine medicine, cfDNA has been explored as a potential biomarker for Equine Recurrent Uveitis (ERU) [8], osteoarthritis, colic, systemic inflammatory response syndrome (SIRS), and sepsis, with varying results [4,5,7,8,9]. In a clinical study, Fingerhut et al. found that cfDNA was increased in serum from horses with ERU compared to healthy controls [8]. In an induced model of equine osteoarthritis, Panizzi et al. found that joint fluid cfDNA, but not plasma cfDNA, was significantly increased at 4 and 9 weeks post-induction, compared to controls [10]. In a prospective clinical study in foals, Colmer et al. found no significant association between cfDNA and culture status, sepsis score, neonatal systemic inflammatory response score (NSIRS), or foal survival. In a prospective clinical study, Bayless et al. showed that compared to healthy horses, plasma cfDNA was elevated in horses presenting for emergency conditions, and specifically in horses with colic. Importantly, in the same study, Bayless et al. showed a matrix effect of equine plasma, including significant autofluorescence and non-specific fluorescent dye binding, interfered with accurate Qubit^TM^ measurement of plasma cfDNA. These authors concluded that accurate fluorescent measurement of plasma cfDNA in adult horses required extraction. Since the Qubit does not numerically calculate for extraction, a correction formula has been developed and validated [5].

The Qubit^TM^ Fluorometer (Invitrogen, Thermo Fisher Scientific) is a compact, tabletop fluorometric device that uses Qubit^TM^ Assays for the quantitation of DNA, RNA, microRNA, and protein, and has been used in numerous cfDNA studies in various species. It is currently unknown whether Qubit measurement of cfDNA in foal plasma is both reliable and accurate. Therefore, the primary objectives of the current study were to investigate foal plasma for potential matrix effect during Qubit measurement of cfDNA, and, depending on results of objective 1, to determine whether neat and/or extracted plasma cfDNA concentrations are significantly different in healthy, sick non-septic (SNS) or septic foals. We hypothesized that matrix effect would interfere with accurate Qubit-measurement of cfDNA in foal plasma. Further, we hypothesized that extracted cfDNA concentrations, and/or cfDNA:neutrophil ratio, would be elevated in septic foals compared to healthy or SNS foals.

## 2. Materials and Methods

For this study, we utilized frozen, banked plasma samples from a previously approved study (North Carolina State University Institutional Animal Care and Use Committee, 20-534). Consent was obtained from all clients at the time of sample collection. Signalment, physical examination, hematology, and biochemistry findings at admission, primary diagnosis, sepsis score [11], and survival to hospital discharge were recorded for all patients. Samples were selected if they met the inclusion criteria of being <7 days of age, had a primary diagnosis, an assigned sepsis score, and were banked during the study period of 2020–2023. Foals were grouped into healthy (clinically healthy, sepsis score ≤ 5 ± negative blood culture), SNS (sepsis score of 6–11 and negative for clinically relevant organisms on blood culture), or septic (sepsis score ≥ 12 ± positive blood culture). Blood culture was not available for all cases.

### 2.1. Sample Collection and Processing

Samples were obtained from the jugular vein of 60 foals (<7 days of age). For hospitalized cases, samples were obtained on admission when foals presented to The University of Iowa, North Carolina State University, or Hagyard Equine Medical Center from the years 2020–2023. Some samples were also collected from healthy foals (<7 days of age) born to the NC State CVM teaching herd. Blood was collected into EDTA vacutainer tubes and immediately plasma was separated by centrifugation at 2300 g at room temperature. Plasma was harvested taking extreme care not to disturb the buffy coat and 1.5 mL aliquots were frozen at −80 °C until thawing and DNA extraction.

### 2.2. CfDNA Measurements on Neat Plasma

Cell-free DNA was measured from thawed plasma using a commercial kit (DNeasy Blood and Tissue Kit; Qiagen, Germantown, MD, USA). 10 µL of plasma was added to 190 µL of Qubit 1× dsDNA HS working solution. Samples were vortexed for 3–5 s and incubated in a dark drawer for two minutes. After incubation, samples were immediately read in triplicate, waiting 1 min between repeat readings of the same tube. Cell-free DNA was determined from the standard curve generated by serial dilution of (*E. coli* bacteriophage) DNA standards. Phosphate buffered saline was used as a negative control.

### 2.3. CfDNA Measurements on Extracted Plasma

DNA was extracted from thawed plasma using a commercial kit (DNeasy Blood and Tissue Kit; Qiagen, Germantown, MD, USA). In brief, 250 µL of plasma was added to a proteinase and buffer mixture and then vortexed. Samples were then incubated at 56 °C for 10 min. After incubating, >96% ethanol was added and mixed. The mixture was then pipetted into a DNeasy spin column placed in a 2 mL collection tube and centrifuged at 6000× *g* for 1 min. This process was repeated an additional two times. After the third centrifugation, the spin column was transferred to a new 1.5 mL microcentrifuge tube and the DNA was eluted by adding 50 µL of Buffer AE to the center of the spin column membrane. After incubation for 5 min, the column was spun a final time and cfDNA was read from the eluted sample as described above.

### 2.4. Cell-Free DNA Measurement

Immediately after processing under each of the above conditions, concentrations of DNA were measured using a compact benchtop fluorometer according to manufacturer instructions (Qubit 4; Invitrogen, Thermo Fisher Scientific, Waltham, MA, USA). The concentration of DNA in each sample was automatically calculated by the fluorometer algorithm based on fluorescence and dilution ratio. The DNA concentration reported by the instrument was converted to cfDNA concentration in the extracted plasma sample using the following equation.
(1)$${\left[\mathrm{cfDNA}\right]}_{\mathrm{plasma}}=\frac{{\left[\mathrm{cfDNA}\right]}_{\mathrm{extracted}\;\mathrm{sample}\;}\times \;\mathrm{elution}\;\mathrm{volume}}{\mathrm{volume}\;\mathrm{of}\;\mathrm{extracted}\;\mathrm{plasma}}$$

### 2.5. Inter-Assay and Intra-Assay Agreement

In brief, intra-assay coefficient of variation (CV) was determined by measuring cfDNA concentrations five times on the same day in six neat plasma samples. Inter-assay CV was determined by measuring cfDNA concentrations in six neat plasma samples on three consecutive days.

### 2.6. Linearity of Dilution

To assess for linearity of dilution, neat and extracted plasma from three foals was individually diluted using molecular biology grade water and the DNase-treated plasma. Dilutions were made as follows: 0:100, 10:90, 20:80, 30:70, 40:60, 50:50, 60:40, 70:30, 80:20, 90:10, 100:0. All dilutions were made to total 30 µL. Cell-free DNA concentrations were determined as in prior steps.

### 2.7. Nonspecific Dye Fluorescence

Plasma samples from 6 healthy foals, 2 SNS foals, and 4 septic foals were treated with Turbo DNase to investigate nonspecific dye fluorescence. Briefly, 250 µL of neat plasma, or 25 µL of extracted plasma, were treated with Turbo DNase (Roche, Basel, Switzerland) that was prepared according to manufacturer instructions. Samples were then incubated at 37 °C for 30 min. Cell-free DNA concentration was then measured as in previous steps. Turbo DNase activity was confirmed by degradation of a known stock DNA sample.

### 2.8. Autofluorescence

To assess the degree of autofluorescence, cfDNA concentration was measured as in previous steps in six plasma samples (two from each foal group), but without the addition of the dsDNA reagent.

### 2.9. cfDNA to Neutrophil Ratio

For all included foals, extracted cfDNA, health status, and neutrophil count were recorded. The following equation was used to determine an extracted cfDNA to neutrophil ratio.
(2)$${\left[\mathrm{cfDNA}\right]}_{\mathrm{plasma}}/\mathrm{Neutrophil}\;\mathrm{Count}\;$$

### 2.10. Data Analysis

Prior to starting the study, sample size calculation was performed based on plasma extracted cfDNA concentrations previously obtained from adult horses. It was determined that we would need to enroll at least 15 foals in each clinical group to detect a minimum difference in means of 3 ng/mL cfDNA, based on a one-way ANOVA with a Type I error rate α = 0.05 and Power, 1 − β = 0.80, with an estimated standard deviation of 5 ng/mL. Data normality was assessed using Shapiro–Wilk test. For non-parametric data, Kruskal–Wallis or Spearman r tests were used. For parametric data, Pearson r test was used. All analyses were performed using GraphPad Prism V 10. Significance was set at *p* < 0.05 or a confidence interval of 95% for all analyses.

## 3. Results

### 3.1. Animals

Samples were collected from a total of 60 foals that met the inclusion criteria for the study between January 2020 and December 2023 (Appendix A). There were 23 (38%) healthy foals with an average age of 36.4 h (range 24–80 h), 17 (28%) septic foals with an average age of 63.5 h (range 24–120 h), and 20 SNS (33%) with an average age of 31.6 h (range 0–240 h). Of the 60 foals, 31 were colts (52%), 26 were fillies (43%), and 3 were not recorded. All foals included in the study were 0–5 days of age. A total of 56/60 (93%) foals survived to discharge (23 healthy, 14 septic, 20 SNS).

### 3.2. Qubit cfDNA Measurement Precision and Accuracy

Coefficient of variation (CV) and linearity of dilution of Qubit measured cfDNA in foal plasma were moderately good to excellent, suggesting reasonable precision and accuracy. Mean intra- and inter-assay CV for measurements of neat plasma cfDNA concentration were 3.4% and 12.95%, respectively (Table 1). Cell-free DNA concentration in dilutions of neat plasma samples demonstrated a high linearity (*p* < 0.0001) (Figure 1). However, there was no significant correlation between extracted and neat plasma cfDNA concentrations in samples from healthy, SNS, or septic foals (Figure 2). Neat and extracted samples from each group of foals were treated with Turbo DNase and cfDNA measurement was repeated. In extracted cfDNA samples, Turbo DNase treatment resulted in 100% elimination of fluorescence, as expected (Appendix A). In neat plasma samples, elimination of fluorescent signal was variable, with DNA signal in most samples degraded by 80–95%, but signal in one sample degraded by as little as 9% (Table 2). Autofluorescence of samples, determined by measuring fluorescence of samples without the addition of the dsDNA dye, was present in all samples and accounted for 22–87% of the cfDNA “concentration”, depending on the sample (Table 2). Together these data show that Qubit fluorescence measurement of cfDNA concentration in neat foal plasma is precise but not necessarily accurate, while Qubit measurement of extracted cfDNA is both precise and accurate.

### 3.3. Plasma cfDNA and cfDNA:Neutrophil Ratios in Healthy, SNS and Septic Foals

Extracted cfDNA concentrations, rather than neat, were used for all group comparisons due to the potential lack of accuracy of cfDNA concentration measurement in neat foal plasma. There were no significant differences in extracted cfDNA concentrations between groups of foals (*p* > 0.05) (Table 3, Figure 3). Further, there was no significant difference in neutrophil numbers (Appendix A) or cfDNA:neutrophil ratios compared between healthy, SNS, and septic foals (Table 4, Figure 4). Five foals (3 healthy and 2 septic) were excluded from neutrophil analysis due to lack of available data.

## 4. Discussion

Cell-free DNA was measured in neat plasma, and following DNA extraction with a commercial kit, using the Qubit 4 fluorometer and 1× dsDNA HS assay kit, from 60 foals. Assay precision and linearity of dilution were moderately good to excellent for neat plasma. Cell-free DNA concentrations in paired neat and extracted plasma were not correlated in healthy, SNS, or septic samples, and individual samples from all groups exhibited varying degrees of non-specific dye binding and autofluoresence when cfDNA was measured in neat plasma. There was no significant difference found in plasma cfDNA concentrations, or cfDNA:neutrophil ratios, between healthy, SNS or septic foal groups. Our data show that matrix effect interferes with accurate Qubit measurement of cfDNA in foal plasma and supports previous findings suggesting that extraction is necessary when evaluating cfDNA concentrations in equine plasma [5]. Our data also support previous findings that plasma cfDNA concentrations are not associated with sepsis diagnosis in foals [9].

Initial analysis of Qubit measured cfDNA concentration in neat foal plasma showed high linearity of dilution and relatively low coefficient of variation (Figure 1, Table 1), consistent with previous reports in humans, dogs, and horses [5,12]. These findings would appear to support fluorescence measurement of cfDNA concentration in neat foal plasma as accurate, precise, and stable over time. However, further investigation showed poor correlation between cfDNA concentrations in neat plasma and extracted samples from both SNS and septic foal groups, suggesting a lack of accuracy of cfDNA measurement in neat plasma samples from sick foals. Given this, samples were further evaluated for non-specific dye binding and autofluorescence.

While the Qubit manufacturer has shown that the dsDNA reagent has excellent specificity, there is the potential for the dsDNA reagent to interact with off-target molecules in plasma. We analyzed 6 foal samples, 2 from each group, for non-specific dye binding by measuring cfDNA concentration (fluorescence) before and after DNA degradation by exogenous Turbo DNase. (Activity of Turbo DNase was confirmed by 100% degradation of signal in extracted samples; Appendix A.) In 5 of the 6 neat plasma samples, % degradation was relatively high, ranging from 86–96%. However, in one sample from a septic foal, % degradation was only 9%, suggesting a significant percentage of the fluorescent signal was coming from either off-target dye binding or autofluorescence. Plasma autofluorescence due to high bilirubin has been previously reported for both dogs and horses [5,13]. Autofluorescence of neat foal plasma was investigated by measuring cfDNA concentration in the absence of dsDNA reagent. Six out of 6 samples, 2 from each group, exhibited autofluorescence that ranged from approximately 23% to 87% of the neat cfDNA measurement.

While there is the potential for variability in efficiency of DNA extraction, the measured DNA concentrations in our starting samples were well within the manufacturer recommendations, and extraction is the standard for human studies analyzing plasma cfDNA [14,15,16]. We did not investigate linearity of dilution in extracted cfDNA samples because there is no matrix that would potentially interfere with cfDNA measurement. We did not investigate coefficient of variation in extracted samples due to the small sample volume created from the extraction process and acknowledge that future investigation with larger sample volumes may be beneficial.

Extraction of cfDNA is not indicated in every species. In dogs, it was concluded that extraction offers no advantage over neat measurement [17]. However, in studies investigating cfDNA in human patients, it is common to extract cfDNA from plasma prior to quantification and additional analysis [15,18,19]. Extraction removes plasma matrix components such as clotting factors and large proteins as well as analytes such as hemoglobin and bilirubin that can contribute to non-specific dye binding and/or autofluorescence. In human medicine, there are efforts to promote cfDNA extraction as a more reliable and easily standardized method of cfDNA analysis across studies [14]. Results from our current study in foals, and our previous study in adult horses, indicate that extraction is necessary for accurate Qubit quantification of cfDNA in equine plasma. Other commonly reported methods of cfDNA quantification include spectrophotometry and plate-based fluorescence assays [20]. Unfortunately, these methods require specialized equipment and would have limited utility as a point-of-care method of cfDNA measurement and were therefore not the focus of this investigation. However, given the limitations of cfDNA measurement with the Qubit, future studies should include investigation of the precision and accuracy of other methods of cfDNA quantification.

Our results show that cfDNA was not significantly different in this population of septic, SNS, and healthy foals, which is somewhat surprising given the previous evidence for cfDNA as a useful biomarker for sepsis in other species. A recent systematic review and metanalysis (18 independent studies) of cfDNA measurement in human patients showed that cfDNA levels were significantly higher in patients with sepsis, and that cfDNA was a useful sepsis biomarker. Specifically, cfDNA level was found to have a pooled sensitivity of 0.81 and specificity of 0.72 for the identification of sepsis in critically ill patients, and a pooled area under the receiver operating characteristic curve (AUC) of 0.76 for prediction of mortality [21]. Cell-free DNA has also been associated with sepsis and systemic inflammation in veterinary species. Dogs presenting for sepsis or trauma had significantly higher levels of plasma cfDNA compared to healthy dogs [12]. Plasma cfDNA was significantly elevated in pigs and mice with experimentally induced sepsis [22]. Plasma cfDNA (extracted) was also higher in adult horses presented for colic compared to healthy horses, and in colic patients with SIRS ≥ 2 compared to non-SIRS colic cases [19]. While the findings of the current study differ from these previous findings on cfDNA, they do corroborate another recent study that also showed no difference in cfDNA in septic, sick non-septic, and healthy foals [9], albeit using neat plasma samples. There are immunological differences between neonatal foals and adult horses that could explain differences between this study and our previous findings [23]. Both phagocytosis and oxidative burst are reduced in healthy foals from birth to 3 months of age [24]. Phagocytic capacity of neutrophils was further decreased in septic and sick non-septic foals compared to healthy foals [25]. To our knowledge, the capacity of foal neutrophils to perform NETosis has not been evaluated. In human infants, several studies have shown decreased or delayed NETosis ability [26,27]. Despite this, markers of NETs, including cfDNA, were found to be increased in human infants with sepsis and levels of NETs were positively correlated with sepsis severity [2]. The same does not appear to hold true for foals. Given the fact that suicidal NETosis is dependent on the oxidative burst [28] and foals less than 3 months old have decreased oxidative burst capacity, it would make sense that NETosis capacity is decreased in foals. Interestingly, one previous study showed that plasma transfusion in septic foals increased neutrophil phagocytosis and oxidative burst, and that serum IgG was positively correlated with oxidative burst [29]. Given that many septic foals present for failure/partial failure of passive transfer of immunity, measurement of cfDNA after plasma transfusion, rather than before, may be more representative of severity of systemic inflammation and sepsis and should be considered as a direction for further research.

Our study found no significant difference in cfDNA:neutrophil ratio between healthy, SNS, and septic foals. This ratio was investigated because neutrophil release of extracelluar traps (NETs) is a prominent source of plasma cfDNA during sepsis, sepsis is known to affect numbers of circulating neutrophils, and low neutrophils (i.e., a degenerative left shift (DLS)) has been identified as a negative prognostic indicator for sepsis in veterinary species. Indeed, Burton et al. found that dogs with a DLS were almost twice as likely to die or be euthanized as diagnosis-matched controls without a DLS [30]. In a separate study, Burton et al. found that cats with a DLS within 24 h of hospital admission were 1.57 times more likely to die or be euthanized than cats without a DLS [31]. Gayle et al. found that foals with neutrophil counts > 4.0 × 10^9^/L had increased odds of survival [32]. Given the evidence associating neutrophil numbers and sepsis severity, we hypothesized that cfDNA relative to neutrophil number might provide a better indicator of neutrophil activation and cfDNA from NETosis than plasma concentration alone. Indeed, an increased cfDNA:neutrophil ratio was associated with non-survival in dogs with sepsis [33]. While we found no significant difference in cfDNA:neutrophil ratio between groups in this study, very few of our cases had low neutrophils and our rate of non-survival was low. Premature foals can also have low white blood cell counts unrelated to sepsis [34], which complicates the comparison of cfDNA:neutrophil ratio as an indicator of sepsis between SNS and septic groups. In future studies, it would be ideal to include cases with a wider range of sepsis severity and outcomes and revisit comparison of absolute and relative cfDNA values.

One limitation of our study was small sample size. Inclusion of more sick and septic foals would allow foals to be grouped by diagnosis to elucidate if infectious conditions (i.e., diarrhea) have higher cfDNA concentrations than non-infectious conditions (i.e., neonatal maladjustment syndrome). For this study, a sepsis score was used to classify foals into their groups. Because of known variability in sepsis score performance across institutions [35] and high rates of false negatives with blood culture [36], it is possible that foals were misclassified, particularly in SNS vs. septic groups. We also elected to include clinically healthy foals with positive blood culture in the healthy group due to the known potential for false positive blood culture in foals [37]. While we felt the most likely cause for positive blood cultures in our healthy foals was contamination during sample collection, transient bacteremia has been reported in healthy foals <12 h of age [37] and could potentially affect innate immune responses such as NETosis. NSIRS is another scoring system available for diagnosis and prognosis of severe illness in neonatal foals. NSIRS has similar reported sensitivity and specificity for sepsis diagnosis in foals as the modified sepsis score used in this study [38]. Although we did not evaluate correlation between NSIRS and cfDNA, Colmer et al. found no association between NSIRS and plasma cfDNA in a population of hospitalized foals, with the caveat that they measured cfDNA concentrations in neat plasma. Another limitation of this study is that only one method of cfDNA measurement, Qubit fluorescence, was utilized. We focused on this form of measurement because of the compact and affordable characteristics of the Qubit device and the potential for stall-side application. In future studies, measurements such as SYTOX green and spectrophotometry should be utilized for comparison. Additionally, cfDNA is only one marker for NETosis and other markers for future research could include myeloperoxidase, CitH3 and neutrophil elastase. Finally, although all foals included in this study were 5 days of age or less, groups were not age matched to the day, and prematurity and gestational age created additional variables potentially impacting the immune function of foals included in this study.

The data from this study offer support for continued investigation of the pathology of NETosis in foals and clinical markers of NETosis in foals, as they appear to differ from adult horses and other species. It is currently unknown whether this difference contributes to the pathophysiology of common diseases in foals such as pneumonia or sepsis. As the extracted plasma cfDNA concentrations from septic foals varied widely, future investigations should target longitudinal cfDNA measurements to determine the impact of timing, stage of sepsis, or response to treatment on cfDNA, and other markers of NETosis, in neonatal foals.

## Figures and Tables

**Figure 1 vetsci-11-00346-f001:**
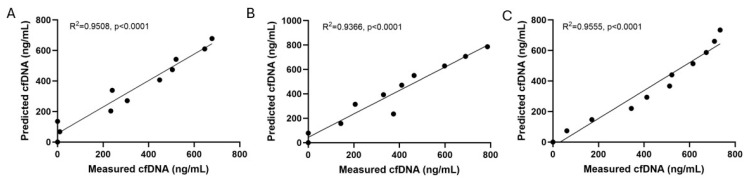
Actual vs. predicted dilutions of Qubit-measured cfDNA in neat foal plasma repeated for three foals. (**A**) Healthy, (**B**) Sick not Septic, (**C**) Septic.

**Figure 2 vetsci-11-00346-f002:**
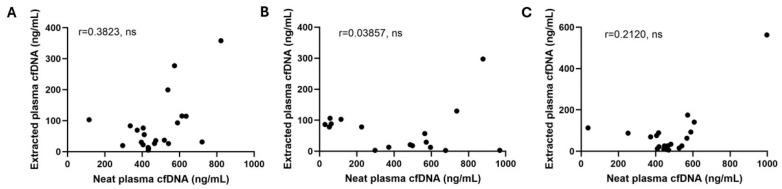
Correlation between neat and extracted cfDNA samples in (**A**) healthy, (**B**) Septic not Sick and (**C**) Septic foals. ns = not significant.

**Figure 3 vetsci-11-00346-f003:**
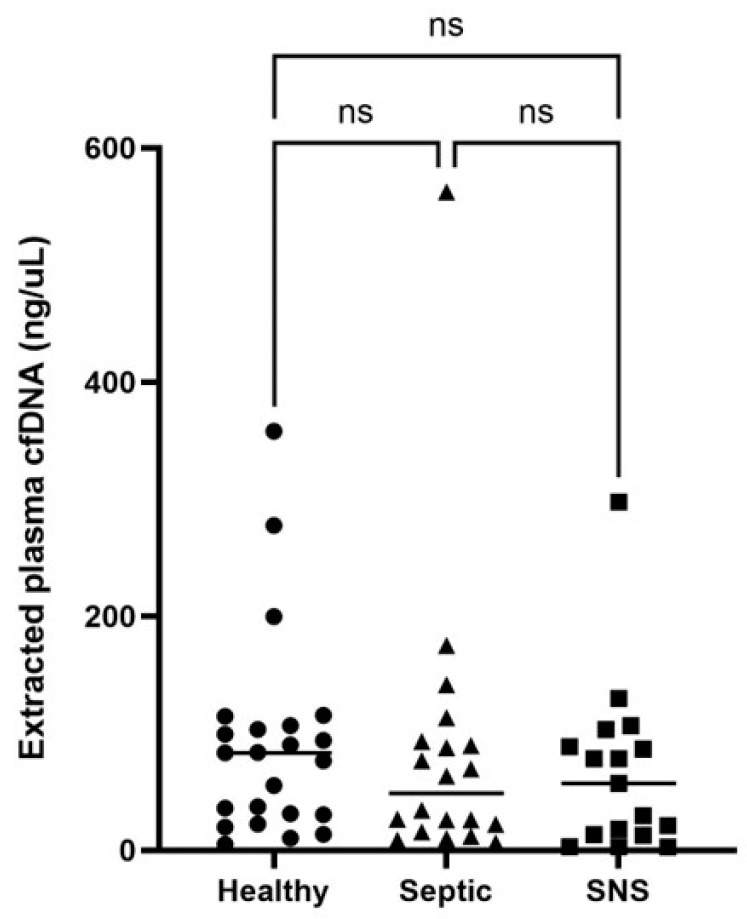
Comparison of extracted cfDNA concentrations of healthy, septic and sick not septic (SNS) foals. Significance set at *p* < 0.05. ns = not significant.

**Figure 4 vetsci-11-00346-f004:**
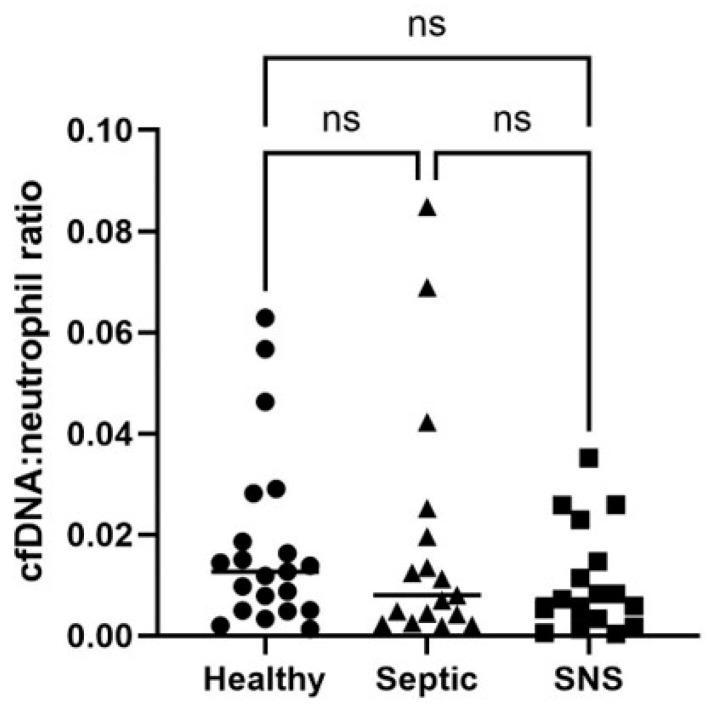
Comparison of cfDNA: neutrophil ratio for healthy, septic and sick not septic (SNS) foals. Significance set at *p* < 0.05. ns = not significant.

**Table 1 vetsci-11-00346-t001:** Intra- and Inter-assay agreements when measured five times during one day and over a three-day period.

Sample	Intra-Assay CV	Inter-Assay CV
1 (SNS)	2.54%	9.55%
2 (Healthy)	4.37%	18.07%
3 (Septic)	1.75%	7.08%
4 (SNS)	2.14%	10.78%
5 (Healthy)	1.88%	15.74%
6 (Healthy)	7.75%	16.50%
Average	3.4%	12.95%

**Table 2 vetsci-11-00346-t002:** Neat plasma cfDNA concentrations before and after treatment with Turbo DNase and autofluorescence as measured with no DNA reagent added. (% degraded = ((neat plasma cfDNA − turbo DNA treated)/neat plasma cfDNA) × 100); (% autofluorescence = (autofluorescence/cfDNA before Turbo DNase) × 100).

Sample	cfDNA before Turbo DNase (ng/mL)	cfDNA after Turbo DNase (ng/mL)	% Degraded	Autofluorescence	% Autofluorescence
1 (Healthy)	720.6	29.8	95.9%	--	--
2 (Healthy)	411	374	9%	195	47.4%
3 (Healthy)	466	--	--	255	54.7%
4 (Healthy)	370	42.4	88.54%	--	--
5 (Healthy)	668	29.8	95.54%	--	--
6 (Healthy)	454	374	17.62%	--	--
7 (SNS)	526.6	32	93.9%	121	22.98%
8 (SNS)	310	43	86.1%	272	87.7%
9 (Septic)	382	42.4	88.9%	255	66.8%
10 (Septic)	875	103	88.2%	332	37.9%
11 (Septic)	492	32	93.50%	--	--
12 (Septic)	568	103	81.87%	--	--
Average	--	--	76.3%	--	52.9%

**Table 3 vetsci-11-00346-t003:** Cell-free DNA concentrations in 60 hospitalized neonatal foals. SS = sepsis score. (Blood culture criteria excluded).

Foal Category	Median (Range) cfDNA (ng/mL)	95% Confidence Interval
Healthy (SS ≤ 5) (*n* = 23)	83.2 (5.28 to 358.1)	31.34 to 103.2
Sick non-septic (SS 6–11) (*n* = 17)	57.2 (2.92 to 297.4)	13.3 to 88.6
Septic (SS ≥ 12) (*n* = 20)	48.73 (6.44 to 562.7)	22.2 to 89.46

**Table 4 vetsci-11-00346-t004:** cfDNA:neutrophil count ratios in 55 hospitalized neonatal foals. SS = sepsis score (Blood culture criteria excluded).

Foal Category	Median (Range) cfDNA:Neutrophil Ratio	95% Confidence Interval
Healthy (SS ≤ 5) (*n* = 21)	0.01265 (0.001272 to 0.06156)	0.004979 to 0.02337
Sick non-septic (SS 6–11) (*n* = 17)	0.007179 (0.0003255 to 0.03511)	0.002522 to 0.01878
Septic (SS ≥ 12) (*n* = 17)	0.008 (0.001844 to 0.08487)	0.003341 to 0.02235

## Data Availability

The original contributions presented in the study are included in the article/Appendix A, further inquiries can be directed to the corresponding author/s.

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
