# Peer review of "Investigation of Extracted Plasma Cell-Free DNA as a Biomarker in Foals with Sepsis"

_vetsci, 2024, doi:10.3390/vetsci11080346_

Round 1

Reviewer 1 Report

Comments and Suggestions for Authors

Dear Authors,

The original paper “Investigation of extracted plasma cell free DNA as a biomarker in foals with sepsis” could add some new information about this potentially new biomarker to identify foals at risk of sepsis but could provide also new insights into the immunological response expected in equine neonates. The paper is well written and quite precise, nevertheless, I have a few remarks and suggestions I would like the authors to consider.

1.      Introduction:

L48-53: Maybe a sentence to explain more what is NETosis and its link with sepsis could be added to clarify the ultimate aim of the study (find a new biomarker or even open research about a potentially new therapeutic target)

L63: It seems to be an error in the cited references: reference 4 (Jackson et al 2019) refers to a study on human trauma patients and should not be cited in equine medicine.

L75: Please define what the “Qubit” is for non initiated people

2.      Materials and methods

L88: please could you precise how the samples and the foals were selected : all the samples stored for a previous study and satisfying the inclusion criteria, ie all foals <7 days of age admitted to the hospital during the years 2020-2023 with available stored frozen plasma at -80°C or only 60 samples were randomly selected in the plasma bank?

L93: Contrary to the study of Colmer et al you have chosen to consider the foal healthy if the sepsis score is low whatever the result of blood culture. Could you please explain this choice into the discussion section

L95: the exclusion criteria are defined as the following: “lack of primary diagnosis, blood culture or sepsis score”, but if I am not wrong, some foals do not have any blood culture results in the supplementary table 1 and should be then excluded from the study (3 foals, one in each group) or the exclusion criteria could be reformulated

L93-94,98: please could you add the timing when the sample was realized for blood culture (upon admission or <24h after admission for example) and if repeated for the same foal the number of samples submitted for blood culture for each foal in your study

L143: please precise the number of samples tested in each group.

L154: if I am not wrong only 55 foals were included for the calculation of cfDNA/neutrophils ratio and in the supplementary table 1, the neutrophil count was not recorded for 4 foals (1healthy and 3 septic), so maybe you could emphasize this point in this section or later in the results.

3.      Results

L180-181: “no significant correlation…from healthy, SNS or septic foals” this sentence seems to be contradicted by the discussion where it is noted “positively correlated in healthy samples but were not correlated in SNS or septic samples” L218-219 even if the correlation coefficient (r 0.3) seems better for healthy foals in figure 2. Please could you explain the different results of correlation between extracted cfDNA and neat plasma cfDNA according to the group in the results section

L212 Table 4: only 55 foals are recorded in the table instead of 56 foals with neutrophil count available in the supplementary material

4.      Discussion

L220: “cfDNA neutrophil ratio”: Even if not significant in the present study, could you explain the interest to calculate cfDNA/neutrophil ratio, how this ratio is expected to vary with sepsis and what is the main advantage to use it ( is it more specific to identify cfDNA generated by NETosis of neutrophils for example) and please add a reference if possible about the use of this ratio in other species.

L218-219 cfr infra (it seems there is a discrepancy between results and discussion sections)

L313-314: “foals were misclassified due to the low sensitivity of sepsis score in foals”. According to Wong et al, 2018, the sensitivity but also the specificity seems to be lower compared to the paper of Brewer in 1988, please add a comment about the choice of this scoring system with the cut-off value of 12 used in this study compared to updated sepsis score, and why you did not use negative blood culture in a way to try to better identify healthy foals.

5.      Supplementary material

Table 1

The primary diagnosis could be added to the table since all foal included in this study  had a primary diagnosis and it could help to interpret sepsis score and to valid a correct classification in the 3 groups (even if it could be difficult to compare the different diagnosis in light of your cfDNA measurements in your population as you mentioned in the discussion)

Could you add an explanation if there a difference between no growth or negative for blood culture (depends if the foal is on antibiotics?).

Some foals have not submitted blood culture and are still included in the study, this could be noted in the material and method section.

Table 3: 55 foals are included instead of 56 in table 1  if I am not wrong

Author Response

Reviewer 1: Thank you for your helpful comments! Below you will find our line by line responses.

1.Introduction:

Comment 1: L48-53: Maybe a sentence to explain more what is NETosis and its link with sepsis could be added to clarify the ultimate aim of the study (find a new biomarker or even open research about a potentially new therapeutic target)

Response 1: Thank you for this comment. The authors have added an explanation of NETosis to Line 52-58, and explained that NETosis is implicated in thrombosis, inflammation and MODS in patients with sepsis.

Comment 2: L63: It seems to be an error in the cited references: reference 4 (Jackson et al 2019) refers to a study on human trauma patients and should not be cited in equine medicine.

Response: 1Thank you for pointing this out. In the preceding paragraph there is discussion of use of cfDNA in human sepsis patients as one of the basis’s for our investigations as humans and horses experience sepsis with may parallel characteristics. This paper discuses both trauma and sepsis and is in line with the proceeding information. We will remove this portion though if the reviewer wishes, but because of the comparative discussion, including examples of research from human papers, we would be pleased to keep this reference.

Comment 3: L75: Please define what the “Qubit” is for non initiated people

Response 2: Thank you. We have added an update that the Qubit is a fluorescence quantification device (Line 76)

  1. Materials and methods

Comment 4: L88: please could you precise how the samples and the foals were selected : all the samples stored for a previous study and satisfying the inclusion criteria, ie all foals <7 days of age admitted to the hospital during the years 2020-2023 with available stored frozen plasma at -80°C or only 60 samples were randomly selected in the plasma bank?

Response 4: Thank you for pointing this out. The manuscript has now been updated to reflect that we included samples of all foals that met our inclusion criteria within the study time frame.

Comment 5: L93: Contrary to the study of Colmer et al you have chosen to consider the foal healthy if the sepsis score is low whatever the result of blood culture. Could you please explain this choice into the discussion section

Response 5: Thank you for your comment. We have added a line to the limitations 324 to address that we did include some foals with a positive blood culture in our healthy group. These positive cultures occurred in healthy foals that were part of a teaching herd. These foals never presented to the hospital for clinical illness and due to the samples being collected on the farm there was a high level of suspicion for contamination. This decision is based on the known possibility of false positive with blood culture, as well as the expertise of one of our authors, Dr. Dembek, who has done extensive research in septic foals.

Comment 6: L95: the exclusion criteria are defined as the following: “lack of primary diagnosis, blood culture or sepsis score”, but if I am not wrong, some foals do not have any blood culture results in the supplementary table 1 and should be then excluded from the study (3 foals, one in each group) or the exclusion criteria could be reformulated

Response 6: Thank you for pointing this out. We have revised this statement to accurately reflect our exclusion criteria. As is consistent with previously published literature and experience of our expert investigator, we chose to include some foals without blood culture results.

Comment 7:L93-94,98: please could you add the timing when the sample was realized for blood culture (upon admission or <24h after admission for example) and if repeated for the same foal the number of samples submitted for blood culture for each foal in your study

Response 7: Thank you for pointing this out. We have updated this paragraph to include “on admission”.

Comment 8: L143: please precise the number of samples tested in each group.

Response 8: The numbers of samples in each group has been updated.

Comment 9: L154: if I am not wrong only 55 foals were included for the calculation of cfDNA/neutrophils ratio and in the supplementary table 1, the neutrophil count was not recorded for 4 foals (1healthy and 3 septic), so maybe you could emphasize this point in this section or later in the results.

Response 9: Thank you for pointing this out. It should be 5 foals (2 healthy and 3 septic). We have added a line about the number of foals included in the data analysis (Line 204)

  1. Results

Comment 10: L180-181: “no significant correlation…from healthy, SNS or septic foals” this sentence seems to be contradicted by the discussion where it is noted “positively correlated in healthy samples but were not correlated in SNS or septic samples” L218-219 even if the correlation coefficient (r 0.3) seems better for healthy foals in figure 2. Please could you explain the different results of correlation between extracted cfDNA and neat plasma cfDNA according to the group in the results section

Comment 10: Thank you for pointing this out. This is a typo in the paper and there should be no significant correlation within any group.

Comment 11: L212 Table 4: only 55 foals are recorded in the table instead of 56 foals with neutrophil count available in the supplementary material

Response 11: The authors have recounted and only count 55 foals. There was a space that has now been removed that may account for the reviewer’s total of 56.

  1. Discussion

Comment 12: L220: “cfDNA neutrophil ratio”: Even if not significant in the present study, could you explain the interest to calculate cfDNA/neutrophil ratio, how this ratio is expected to vary with sepsis and what is the main advantage to use it ( is it more specific to identify cfDNA generated by NETosis of neutrophils for example) and please add a reference if possible about the use of this ratio in other species.

Thank you for this suggestion. We have added to the discussion on this topic starting at line 331.

Comment 13: L218-219 cfr infra (it seems there is a discrepancy between results and discussion sections)

Response: Thank you. Good catch this was a typo and has not been corrected to reflect no correlation between any of our sample groups.

Comment 14: L313-314: “foals were misclassified due to the low sensitivity of sepsis score in foals”. According to Wong et al, 2018, the sensitivity but also the specificity seems to be lower compared to the paper of Brewer in 1988, please add a comment about the choice of this scoring system with the cut-off value of 12 used in this study compared to updated sepsis score, and why you did not use negative blood culture in a way to try to better identify healthy foals.

Thank you for this question. We agree that this is challenging. While new scoring systems have been reported, to our knowledge there is not a significant difference between performance of the new and old scoring systems. Given this, we felt that the scoring system used is familiar to the audience who will read this research, is clinically relevant and is as good an option as any at this point. Validation of a new scoring system was beyond the scope of this project, but we are excited that discovery of new sepsis biomarkers could present an opportunity to revise and improve foal sepsis scores in the future.

Regarding blood culture, given the known potential for false positives from human error or transient bacteremia in otherwise healthy foals, we did not want to eliminate cases of foals that were clearly clinically healthy, on the basis of a single test.

  1. Supplementary material

Table 1

Comment 15: The primary diagnosis could be added to the table since all foal included in this study had a primary diagnosis and it could help to interpret sepsis score and to valid a correct classification in the 3 groups (even if it could be difficult to compare the different diagnosis in light of your cfDNA measurements in your population as you mentioned in the discussion)

Response 15: Thank you for this suggestion. The primary diagnosis has been added to the supplementary material.

Comment 16: Could you add an explanation if there a difference between no growth or negative for blood culture (depends if the foal is on antibiotics?).

Response 16: This has been updated all should say Negative

Comment 17: Some foals have not submitted blood culture and are still included in the study, this could be noted in the material and method section.

Response 17: Thank you for pointing this out. The inclusion criteria has been updated to include foals with and without a blood culture.

Comment 18: Table 3: 55 foals are included instead of 56 in table 1  if I am not wrong

Response 18: The authors only count 55 foals. There was an extra space that has since been removed.

Reviewer 2 Report

Comments and Suggestions for Authors

Cell-free DNA (cfDNA) can be released in various bodily fluids, including blood plasma, urine, and saliva from normal cell turnover, apoptosis, necrosis, or from specific pathological conditions. It is considered a valuable biomarker with broad applications in human and animal medical diagnostics and monitoring to provide critical information in a minimally invasive manner. However, it is currently unknown whether fluorescent measurement of cfDNA in foal plasma is both reliable and accurate. The authors of this paper investigated foal plasma for potential matrix effect during fluorescent measurement of cfDNA. The authors sought to determine whether neat and/or extracted plasma cfDNA concentrations are significantly different in healthy, sick non-septic (SNS) or septic foals. The found that in neat plasma, the matrix effect interferes with accurate Qubit measurement of cfDNA which supports previous findings. Therefore, extraction is necessary when evaluating cfDNA concentrations in equine plasma. In addition, their data also support previous findings that plasma cfDNA concentrations are not associated with sepsis diagnosis in foals.

Comments:

·  The authors found that cfDNA was not significantly different in the population of septic, sick non-septic and healthy foals. Since it contradicts previous published evidence for cfDNA as a useful biomarker for sepsis in other species, it would be useful for the authors conducted experiments to investigate why that was the case. There were some discussions for possible reasons to make their argument stronger, those could be tested. For example, comparison of phagocytosis and oxidative burst capability of cells in foals in septic, sick non-septic and healthy foals. Or inclusion of other methods to measure NETosis.

·      Some minor edits, spacing, etc. which may be fixed during final editorial stage.

Author Response

Thank you for your feedback. The authors agree that investigations of phagocytosis and ROS capability would be an excellent idea for future research, but are beyond the scope of the current project. We are hopeful that our current results provide excellent justification for grant applications to continue our investigation into species differences regarding NETosis and sepsis.

We also agree that the addition of complementary methods to measure NETosis, such as ELISA's for MPO, could provide additional information and the lack of additional methods is discussed as a limitation regarding specificity for NETs. We chose to focus on the Qubit for this project because of its use in previous publications and its potential for use as a “stall-side” test. We think that the main interest of our results lies with the evidence for matrix effect in neonatal plasma, which is consistent with our previous findings in adult horse plasma, and the lack of correlation between cfDNA and neonatal sepsis once the matrix effect is removed. This is consistent with another previous study, but the previous study did not investigate matrix effect, which is why our manuscript adds important new information to the existing literature.

Reviewer 3 Report

Comments and Suggestions for Authors

Overall this is a nice techniques paper and is well-written and presented. I have a few queries and concerns that would need to be addressed.

There are multiple small grammatical and punctuation errors that need to be addressed, this of course will be the job of the copy editors.

Although commonly used, the definitions of SNS and Septic groups is probably not a real one. There is good possibility that there are both false negatives and false positive using 'and/or blood culture positive' as a criteria for classification. It is known that normal foals can be transiently blood culture positive in the first few days of life without ever developing illness. The accuracy of sepsis scores across institutions is also poor due to institutional variability. This has been demonstrated in many studies at this point. While the system the authors have used is common, the potential flaws of the classification need to be pointed out.

Results:

Please give the survival status within all 3 groups, not just overall. Mean/median ages plus a range of foals in each classification group should be stated. Age impacts many things in the first few days of life and detection of infection is easier in the slightly older foals.

Table 1: why was this not done with a 'septic' foal sample. Ideally 2 foals from each health status group should have been used here. Or, simply done in healthy foals only and stated as such with reasons for it.

Figure 1: please give the classification status of the 3 foals used here in the figure legend, . For all analyses, the health classification of foals used needs to be stated, and it would have been preferable that all 3 health status groups were represented.

Figure 2, please redo the figures so that the horizontal axis is the same between all groups.

Discussion

Lines 237-249- Refer to the results specific section for these statement for the ease of the reader, please.

Line 255-257: Do you suggest this be done in future with appropriate volumes collected and processed?

Lines 269-271: Good point, POC would be ideal....and is the way of the future in sepsis diagnostics.

Comments on the Quality of English Language

Language is excellent, just a few grammatical and punctuation errors.

Author Response

Reviewer 3: Thank you so much for your helpful comments! Below you will find our line by line responses. 

Comment: 1 Although commonly used, the definitions of SNS and Septic groups is probably not a real one. There is good possibility that there are both false negatives and false positive using 'and/or blood culture positive' as a criteria for classification. It is known that normal foals can be transiently blood culture positive in the first few days of life without ever developing illness. The accuracy of sepsis scores across institutions is also poor due to institutional variability. This has been demonstrated in many studies at this point. While the system the authors have used is common, the potential flaws of the classification need to be pointed out.

Response 1: Thank you for this comment. The authors agree that the classification of sepsis is challenging and prone to error for the reasons the reviewer points out. We have added additional information to the discussion (Line 324) address the blood culture and also variability across institutions.

 Results:

Comment 2: Please give the survival status within all 3 groups, not just overall. Mean/median ages plus a range of foals in each classification group should be stated. Age impacts many things in the first few days of life and detection of infection is easier in the slightly older foals.

Response: 2: Thank you for pointing this out. This information as now been added to Line 175-181, the results under animals section.

Comment 3: Table 1: why was this not done with a 'septic' foal sample. Ideally 2 foals from each health status group should have been used here. Or, simply done in healthy foals only and stated as such with reasons for it.

Response 3: Thank you for identifying this typo in the paper. When the files were cross referenced, foal 3 which is foal 20-10 in the supplementary and foal 4 which is 20-06 in the supplementary were mislabeled in this table. We agree ideally two foals from each health status should have been included but we had limited access to septic samples.

Comment 4: Figure 1: please give the classification status of the 3 foals used here in the figure legend, . For all analyses, the health classification of foals used needs to be stated, and it would have been preferable that all 3 health status groups were represented.

Response 4: Thank you for pointing this out. This has now been updated.

Comment 5: Figure 2, please redo the figures so that the horizontal axis is the same between all groups.

Thank you for this comment. The figure has now been updated.

Discussion

 Comment 6: Lines 237-249- Refer to the results specific section for these statement for the ease of the reader, please.

Comment 6: Thank you for pointing this out and the Figure and Table number have been added.

Comment 7: Line 255-257: Do you suggest this be done in future with appropriate volumes collected and processed?

Response 7: We do and have added an acknowledgement of this. It takes around 400 ul of neat plasma sample in order to extract 10-20 ul of sample, so design of an IACUC that allows much larger blood sample collection may be useful for further research.

Comment 8: Lines 269-271: Good point, POC would be ideal....and is the way of the future in sepsis diagnostics.

Response 8: Thank you! The authors completely agree with this statement.

Round 2

Reviewer 2 Report

Comments and Suggestions for Authors

The paper is a acceptable in its current form.